# Marine Antibiofouling Properties of TiO₂ and Ti-Cu-O Films Deposited by Aerosol-Assisted Chemical Vapor Deposition

**Caroline Villardi de Oliveira** [1,2,3], **Julie Petitbois** [4], **Fabienne Faÿ** [4], **Frédéric Sanchette** [1,2], **Frédéric Schuster** [2,5], **Akram Alhussein** [1,2], **Odette Chaix-Pluchery** [3], **Jean-Luc Deschanvres** [3] and **Carmen Jiménez** [3,*]

1. ICD-LASMIS, UTT, Antenne de Nogent, Pôle Technologique Sud Champagne, 52800 Nogent, France; carolvillardi@gmail.com (C.V.d.O.); frederic.sanchette@utt.fr (F.S.); akram.alhussein@utt.fr (A.A.)
2. NICCI, LRC CEA-ICD LASMIS, UTT, Antenne de Nogent, Pôle Technologique Sud Champagne, 52800 Nogent, France; frederic.schuster@cea.fr
3. Univ. Grenoble Alpes, CNRS, Grenoble INP, LMGP, 38000 Grenoble, France; Odette.chaix@grenoble-inp.fr (O.C.-P.); jean-luc.deschanvres@grenoble-inp.fr (J.-L.D.)
4. Laboratoire de Biotechnologie et Chimie Marines (LBCM), UE3884, Université de Bretagne Sud (UBS), Université Européenne de Bretagne (UEB), BP92116, CEDEX, 56321 Lorient, France; julie.petitbois@univ-ubs.fr (J.P.); fabienne.fay@univ-ubs.fr (F.F.)
5. CEA, PTCMP, Centre de Saclay, 91191 Gif-sur-Yvette, France
* Correspondence: Carmen.jimenez@grenoble-inp.fr; Tel.: +33-456-529-334

**Abstract:** The actual interest in developing light-induced catalytic coatings to act as an antibiofouling alternative has recently prompted interest in the incorporation of Cu into TiO₂ films, working as a visible light sensitizer catalyst. TiO₂ and new Ti-Cu-O films with Cu contents ranging between 16% and 75% Cu/(Cu + Ti) are deposited by aerosol-assisted metalorganic chemical vapor deposition at a substrate temperature of 550 °C. The films are composed of TiO₂ anatase phase, mixed with Cu₂O when including Cu in the composition. Pure TiO₂ films' morphologies are characterized by the formation of microflower-like structures with nanometric petals, which induce a high specific surface. These features are not present in Ti-Cu-O films. A UV-Visible study revealed that the optical band gap energy decreases with increasing Cu content. Interestingly, Ti-Cu-O films presented a highly photo-catalytic activity in the orange-G degradation. Marine biofouling field tests in Lorient's Harbor in France and in vitro tests were carried out in order to evaluate the antifouling performance of the films, revealing that topography and chemical composition can act differently on different species. Field tests revealed that TiO₂ microflowers reduced the fouling coverage. Besides, Ti-Cu-O films with 16 at.% Cu presented lower fouling coverage than films containing 58 at.% Cu. In vitro tests using two diatoms (*P. tricornutum* and *N. perminuta*) showed that the spaces between microflowers play a significant role in the adhesion of diatoms: microalgae adhere less when spaces are bigger than their cells, compared to when spaces are of the same size as cells. Films containing Cu did not alter *N. perminuta* growth nor adhesion, while they affected *P. tricornutum* by lowering its growth rate and adhesion without noticeable toxicity. Indeed, Cu-Ti-O is a very promising non-toxic fouling release film for marine and industrial applications.

**Keywords:** TiO₂; copper; antibiofouling; Aerosol-CVD; diatoms

## 1. Introduction

Photocatalytic $TiO_2$ films can be applied as self-cleaning surfaces to combat fouling problems in many types of environments [1,2]. Biofouling, defined as the accumulation of microorganisms (bacteria, fungi, microalgae, etc.) and macroorganisms (macroalgae, mollusks, etc.) on wetted surfaces, can cause mechanical problems in a wide range of industrial processes [3]. It has been demonstrated that photocatalytic activity could induce an antifouling activity. Indeed, under UV irradiation, the generation of Reactive Oxygen Species is promoted, which damages the biological cell membrane and decomposes the organic matters present on the surfaces [4–6]. Additionally, virus inactivation by photocatalytic $TiO_2$ might occur through the generation of $O_2^-$ and *OH followed by damage of the viral protein and genome [7].

Inactivating viruses, bacteria and macroorganisms by photocatalysis is safe, cost-effective, fast and permanent. The requirements for being an efficient visible light-active photocatalyst are having high conductivity, long-charge lifetime, direct pathways to carry photogenerated charge, good chemical stability in water, and, at the same time, high surface area for the adsorption of targeted species. $TiO_2$ has been extensively studied [8], and it still is nowadays [9–12], because of its photocatalytical activity, combined with other suitable properties. In order to improve and control the structural, optical and electrical properties of $TiO_2$ and increase the photocatalytic properties, the incorporation of transition metals (Mn, Ni, Cu, Fe) has been studied [11,13–15]. Besides, mixed phase or doped materials present higher charge separation at the interfaces, leading to higher photocatalytic activities, compared to visible light-active photocatalysts [16,17].

Copper has been proposed as a doping element for $TiO_2$ to enhance its functional properties [15]. Zhang et al. [18] predicted that the substitution of $Ti^{4+}$ by $Cu^{2+}/Cu^{1+}$ in the $TiO_2$ lattice induces the creation of oxygen vacancies and additional impurity bands right above the valence band maximum of pure $TiO_2$, which results in a significant reduction of both band gap and the recombination rate of photoelectron–hole pair, improving the photocatalytic performance and visible light absorption of $TiO_2$. The combination of Cu and $TiO_2$ has also been tested as an antibacterial and antibiofouling alternative. Vishwakarma et al. [19] synthesized a film based on nanocrystalline copper over titanium coupons, achieving biofouling resistance to micro- and macroorganisms. Rtimi and Kiwi [20] reported nanocomposites of Cu mixed with $TiO_2$ over textiles and polymers, achieving bacterial inactivation. Moreover, they explained how Cu shows multiple antibacterial mechanisms depending on bacteria nature and environmental conditions.

Physical modification of the surface topography is also a strategy for fighting against biofouling in the health and naval industries. Engineered surface topographies with special geometries of ordered features have been shown to inhibit microorganisms' adhesion. One of the most potent micro-patterns is the Sharklet one, a shark skin-inspired ordered topography [21]. Carve et al. [22] reviewed 75 articles about engineered surface topographies as potential sources of new non-toxic antifouling surfaces. It was shown that successful textures consisted almost exclusively of regularly arranged geometric features, in contrast with irregular textures which tended to increase fouling. In particular, the attachment point theory establishes that topography scale, such as spaces between features, influences the adhesion of microorganisms [23]. Schumacher et al. demonstrated that the more the organism can move freely on a surface, the more it becomes difficult for it to decide on the most appropriate place to settle. However, studies generally focus on a few phyla at the same time when testing the efficiency of surfaces. Several studies have proven that the presence of mixed scale on hierarchical structures, meaning irregular surfaces, might deter foulers from various phyla to settle [23].

In this work, the influence of the incorporation of high contents of Cu into $TiO_2$ thin films on structural, optical and photocatalytic activity is first studied. The deposition by Aerosol-Assisted Metal Organic Chemical Vapor Deposition (AAMOCVD) of Ti-Cu-O films allows for the addition of very high concentrations of Cu into $TiO_2$, a new approach not studied until now. The antifouling properties of these $TiO_2$ and Ti-Cu-O thin films are evaluated as a function of two characteristics: the $TiO_2$ film topography presenting flower-like microstructures, and the Cu content in Ti-Cu-O films.

In vitro and in situ tests were used to evaluate the biofouling and toxicity of Ti-Cu-O films, which is of real interest in assessing the efficiency of these new thin films. Results show the high potential of the proposed coatings: the presence of microflowers as well as copper on the surface of the films controls the adhesion of biofouling without toxicity towards planktonic organisms. This approach is particularly employed due to regulatory constraints. Our approach is not based on a release of biocides.

## 2. Materials and Methods

### 2.1. Film Growth

TiO$_2$ and Ti-Cu-O film depositions were performed by Aerosol-Assisted Metal Organic Chemical Vapor Deposition (AAMOCVD) using a home-made reactor working at atmospheric pressure. The substrates used for depositing copper-titanium oxide films were (100)-oriented silicon single crystal and alkaline earth boroaluminosilicate glass (Corning C1737). Substrates of 5 cm$^2$ × 5 cm$^2$ were used, but for some experiments they were cut with a diamond tip into squares with size of about 1.5 cm$^2$ × 1.5 cm$^2$. Previously to deposition, substrates were washed in an ethanol solution with ultrasound for 10 min, rinsed with deionized water and dried with N$_2$.

TiO$_2$ films were grown using Titanium (IV) oxide bisacetylacetonate (C$_{10}$H$_{14}$O$_5$Ti, Strem Chemicals, Newburyport, MA, USA) as precursor dissolved in ethanol [24]. Precursor solutions were prepared with two different titanium concentrations of 0.03 and 0.06 M, which allow for controlling the density of microflowers, to grow the samples used for the in vitro tests.

The copper precursor was Cu(II) acetylacetonate (Cu(AcAc)$_2$) (98%, STREM Chemicals), also dissolved in ethanol, already used in Cu$_2$O film-deposition by AACVD [25]. Two starting solutions with cationic concentrations of Cu 0.01 M and 0.03 M were first prepared. Since Cu(AcAc)$_2$ has a low solubility in alcohol solvents, 0.01 M ethylene diamine (99%, Sigma Aldrich, St. Louis, MO, USA) was added to increase the solubility. The solution was stirred during heating at 50 °C for 15 h to obtain a homogeneous solution. The copper solution was then mixed with the titanium solution with a concentration of 0.03 M in order to obtain two mixed solutions, one with [Cu]/[Ti] = 1/3 and another with [Cu]/[Ti] = 1. The mixture was stirred until complete mixing at room temperature. The optimization of TiO$_2$ film deposition was already presented in [26]; we used the same deposition parameter as this study for TiO$_2$ and Cu-Ti-O films, mainly characterized by a deposition temperature of 550 °C for 40 min. A carrier gas flow of 6 L·min$^{-1}$ and a solution feeding rate of 3 mL·min$^{-1}$ were used. For more details about deposition conditions, see [26].

### 2.2. Film Characterizations

Film morphology and thickness were observed by Field Emission Scanning Electron Microscopy (FESEM) conducted in a FEI Quanta 250 field emission scanning electron microscope for high-resolution images. The films thicknesses were determined from SEM images of the cross section of films deposited on silicon substrates. The samples prepared in this way (surface sections cut out of the films) were mounted on SEM-specimen holders by means of a conductive carbon-adhesive. The chemical composition of films was obtained by energy-dispersive X-ray spectrometry (EDS) microanalysis using an SDD Bruker AXS-30 mm$^2$ detector (FESEM, Hillsboro, OR, USA). Atomic force microscopy (AFM) was performed in tapping mode using a Veeco D3100. Several areas of 5 μm × 5 μm were scanned to evaluate the roughness of the samples.

The crystallinity and phase identification of the samples were both analyzed by X-ray diffraction (XRD) using a Bruker D8 Advance diffractometer equipped with a monochromatic Cu K$\alpha$1 radiation ($\lambda$ = 0.15406 nm) (Billerica, MA, USA). The $\theta$–2$\theta$ scanning mode using a Bragg–Brentano configuration was performed between 20° and 80° at 0.011° intervals with an acquisition time of 2 s, and the Grazing Incidence X-ray Diffraction (GIXRD) patterns were acquired in the same range with incidence and exit angles maintained around 0.5°.

Raman spectra were collected using a Jobin-Yvon/Horiba LabRam multichannel spectrometer equipped with a liquid $N_2$-cooled charge-coupled device (CCD) detector. Experiments were conducted in the micro-Raman mode at room temperature in a backscattering geometry. The 514.5 nm line of an $Ar^+$ laser was used as the excitation line. The laser power at the sample surface was 66 µW with a spot size of about 1 µm. The recorded spectra were calibrated using Si spectra at room temperature. The integration times were adjusted in order to obtain a high signal-to-noise ratio.

UV-Visible-Near Infrared spectrometry was carried out in a Lambda 950 spectrophotometer from Perkin Elmer in the range of 250 nm to 2500 nm working in total transmission mode and using an integrating sphere.

### 2.3. Photocatalytic Activity

The photocatalytic activity of $TiO_2$ films was evaluated by measuring the decomposition rate of aqueous Orange G solutions ($C = 5 \times 10^{-5}$ M) using a UV–VIS Perkin Elmer Lambda 35 performing the same protocol explained in our previous study [26].

### 2.4. Marine Biofouling Field Tests

Selected samples were tested for antifouling activity in field tests. In total, 12 coupons of each kind of coating were pasted on polycarbonate panels, which were stuck onto a metallic support and immersed in a natural marine environment, in Kernevel Harbour (Lorient, France, Atlantic Coast), in April 2019, at a depth of 50 cm. The coupon of each coating was sampled after 10, 17, 25 and 38 days, and cleaned by wiping back and forth in natural seawater to discard weakly attached and residual suspended cells and debris. Samples for SEM observations were treated according to the protocol of Faÿ et al. [27] and observed with a JEOL JSM 6460 LV scanning electron microscope (Mundelein, IL, USA). Images were taken under an acceleration voltage of 3 keV, from secondary electrons. Cells on the pictures were colorized and processed by ImageJ software to estimate the percentage covering rate (colorized pixels over pixels of background).

### 2.5. Marine Biofouling In Vitro Tests

#### 2.5.1. Strains and Culture Conditions

As diatoms represent the major colonizers in our field tests, two organisms of this phylum were used as models in our in vitro tests. *Phaeodactylum tricornutum* (AC590) was obtained from Algobank (Biological Resource Center of the University of Caen Normandie, Caen, France) and *Navicula perminuta* (CCAP No. 1050/15) from the Scottish Marine Institute (The Scottish Association for Marine Science, Oban, UK). *P. tricornutum* is 6.3 ± 0.5 µm long and 2.3 ± 0.2 µm wide (mean ± SD) (n = 10) and *N. perminuta* is 8.5 ± 1.7 µm long and 3.0 ± 0.3 µm wide (mean ± SD) (n = 10).

Both diatoms were cultured in 250-mL Erlenmeyer flasks containing 100 mL of sterile artificial seawater (ASW, 30 g/L) supplemented with Guillard's F/2 medium at 2% (Guillard and Ryther, 1962, Sigma Aldrich), and treated with a mixture of three antibiotics for axenization: chloramphenicol (100 µg/mL), penicillin G (1000 µg/mL) and streptomycin (500 µg/mL) [28]. They were maintained at 20 °C under continuous illumination by cool white fluorescence lamps of 110 µmol·m$^{-2}$·s$^{-1}$ photons in 18h:6h light:dark periods in a Hélios 600 phytotron (Cryotec, Saint-Gély-du-Fesc, France). Cultures were initiated at approximately $10^3$ cells·mL$^{-1}$. After 2 weeks, both strains were in the stationary phase. Before use, cultures were synchronized [29]. At the beginning of the tests, all cells started from the G2 phase of the cell cycle, favoring reproducible results.

#### 2.5.2. Adhesion Assay in Static Conditions

Cell concentration in the culture flask was estimated via the Malassez counting chamber method. A volume containing approximately $10^6$ cells was harvested in the 250-mL Erlenmeyer flask and diluted with culture medium to obtain $10^5$ cells·mL$^{-1}$ in 10 mL for attachment experiments.

Test panels (approximately 2.25 cm$^2$, n = 3) were individually immersed horizontally in a Petri dish (6-cm diameter) containing 10 mL of ASW with 2% Guillard's F/2 medium. After 72-h adhesion in the same culture conditions as explained previously, the test panels were gently rinsed by wiping back and forth in sterile ASW. This process allowed weakly attached and residual suspended cells to be removed. Thanks to their natural autofluorescence due to their chlorophyll pigments ($\lambda_{excitation}$ = 633 nm, $\lambda_{emission}$ = 638–720 nm), adhered microalgae were observed by confocal laser-scanning microscopy (CLSM) on an area of 0.5 mm$^2$. Images (six per test panel) were acquired by Zen Software (Zeiss) and then processed by ImageJ software to estimate the covering rate (pixels due to autofluorescence over pixels of background) in percentages. Glass is used as a standard to compare between different batches and ensure the good operation of the experiments.

### 2.5.3. Toxicity Assay

The toxicity of the surfaces towards planktonic and adhered microalgal cells (72 h of adhesion) was evaluated by Sytox Green staining. Planktonic cells were harvested and concentrated by centrifugation (3000 rpm for 10 min at 4 °C). They were suspended in 500 µL of sterile ASW. One hundred microliters of this suspension were mixed with 100 µL of an aqueous solution of Sytox Green (final concentration: 0.5 µM) in a well of a 96-well microplate. After 30 min of incubation at 20 °C in the dark, the fluorescences of total cells ($\lambda_{excitation}$ = 633 nm, $\lambda_{emission}$ = 680 nm) and stained cells ($\lambda_{excitation}$ = 485 nm, $\lambda_{emission}$ = 535 nm) were measured using a TECAN microplate reader (Magellan, France) [30]. The percentage of dead cells was calculated as (fluorescence of dead cells/fluorescence of total cells) × 100. The negative control depended on the cells in contact with glass only. Positive control depended on the cells that underwent a treatment with microwaves (3 × 10 s).

Adhered cells were stained with an aqueous Sytox Green solution in the same conditions as above by covering test panels with 500 µL of solution. Adhered dead cells were observed by CLSM on an area of 0.5 mm$^2$ ($\lambda_{excitation}$ = 485 nm, $\lambda_{emission}$ = 500–555 nm). Images processing was performed as described previously.

### 2.5.4. Statistical Analysis

All data were analyzed by IBM SPSS Statistics version 25. We have confirmed that they were normally distributed with homogeneous variances. Significant differences between groups were determined by the *t*-test. All data presented were mean ± standard deviation (SD) and the significance level of 0.05 was applied. Graphs were drawn with GraphPad Prism 8.2.1 with *p*-values of *t*-tests represented as: *: $p \leq 0.05$; **: $p \leq 0.01$; ***: $p \leq 0.001$; ****: $p \leq 0.0001$.

## 3. Results and Discussion

(a)　Pure TiO$_2$ layers

In a previous study, TiO$_2$ films containing flower-like microstructures were grown using the same deposition parameters. It has been previously proven that these microstructures are reached only when working with a very high precursor flow rate (3 mL·min$^{-1}$, meaning 0.027 mol·min$^{-1}$ of TiO(acac)$_2$) in a narrow deposition temperature window, probably generating supersaturation CVD conditions [24]. The photocatalytic activity of these TiO$_2$ films was found to be very high, probably due to the high specific surface generated by the nanometric hierarchical flower-like structures [26]. The increase in titanium concentration in the solution (0.06 M) resulted in the increase in the microflowers' density.

Nevertheless, in these deposition conditions, the thickness homogeneity on the 5 × 5 cm$^2$ surface was not optimal. Samples deposited with a titanium concentration of 0.03 M were cut into two parts as a function of the presence of microflowers or not. Samples with microflowers were labeled TiO$_2$_3 WF, and other ones without microflowers TiO$_2$_3 WOF. Samples obtained from a solution with a Ti concentration of 0.06 M presented a higher microflowers density, and were labeled TiO$_2$_6 WF. These samples were specifically grown to evaluate the influence of topography in the diatom tests.

(b)   Ti-Cu-O

As previously explained, the deposition conditions inducing the microflower structure led to inhomogeneous thicknesses. Nevertheless, we preferred to keep these deposition conditions for the growth of Ti-Cu-O films to ensure their comparison with $TiO_2$ films.

The atomic composition was measured by EDS for films deposited on silicon wafers and Corning glass substrates using the two different solutions. When probing different points of the sample, it was found that cationic composition, measured as at.% Cu, was reproducible across the whole surface, even for different thicknesses.

When using a solution with a cationic ration of [Cu]/[Ti] = 1/3, i.e., 25 at.% Cu, the cationic content in the deposited film was found to be 16 at.% Cu. When working with a solution [Cu]/[Ti] = 1, i.e., 50 at.% Cu as cationic content in the solution, the dispersion of the composition was high and varied from one sample to another, with values ranging between 50 and 60 at.% Cu for films deposited on glass and 58 to 75 at.% Cu for films deposited on silicon. The main difference between the two solutions was the total cationic concentration; 0.02 M for the first one and 0.03 M for the second one. We found a dispersion in film composition from run to run, probably due to the copper diffusion in the vapor phase when using a high flow rate for the carrier gas. It is important to know that when depositing on a solution containing only copper at the same conditions, no film was grown. $Cu_2O$ deposition was only possible at a lower deposition temperature (300–350 °C) [25].

From now, Ti-Cu-O films will be labeled XTiCuO, where X is the atomic percentage of copper measured by EDS.

### 3.1. Morphology of TiO$_2$ and Ti-Cu-O Films

An average thickness was calculated using cross section SEM images for all the films deposited on silicon. All films presented a thickness value of 400 nm, whatever the copper content, except the pure $TiO_2$ film which presented a thickness of 300 nm. The deposition rate was around 7.5 nm/min for $TiO_2$ and 10 nm/min for Ti-Cu-O films.

$TiO_2$ and Ti-Cu-O films deposited on glass were observed by SEM, and the microstructures are shown in Figure 1 at the two magnifications. Figure 1a presents pure $TiO_2$ films containing microflowers grown from a solution with a titanium concentration of 0.03 M; this microstructure was already presented in our previous study [24]. Figure 1b corresponds to the deposition of pure $TiO_2$ films using a higher titanium concentration of 0.06 M. The increase of microflower density is mainly noticeable at lower magnifications. The morphology of films containing copper is rather different because of the absence of microflowers. Increasing the Cu content led to a granular and rough morphology, as shown in Figure 1.

Ti-Cu-O films deposited on Corning glass substrates were also characterized by AFM, which allowed us to measure the roughness through the root-mean-square (RMS) of height deviation. This quantitative technique agrees with the SEM observation indicating higher roughness with the increase in Cu content. It is important to say that $TiO_2\_3$ WF and $TiO_2\_6$ WF were not characterized by AFM because their microflower structures were too high for the availability of z displacement. We previously measured these microstructures by a profilometer, and the height values of 7 μm were obtained [20].

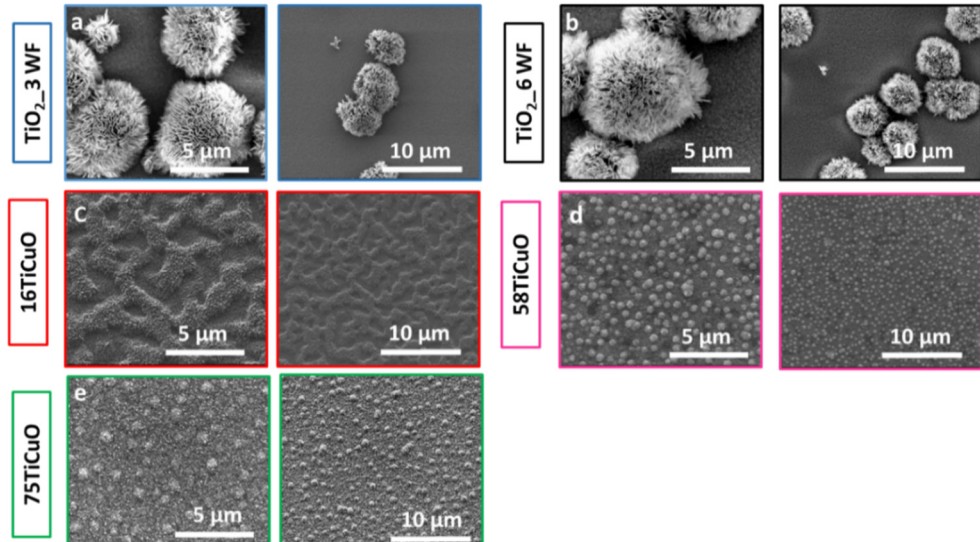

**Figure 1.** SEM images of microstructures corresponding to films deposited by AA-MOCVD on Corning glass substrates: (**a**) the surface of a TiO₂ layer obtained from a solution 0.03 M of titanium and containing microflowers, sample TiO₂_3 WF; (**b**) the surface of a TiO₂ layer obtained from a solution 0.06M of titanium and containing microflowers, sample TiO₂_6 WF; (**c**) surface of the sample 16TiCuO, Ti-Cu-O layer grown from a solution with a cationic ration Cu/Ti = 1/3; surface of samples (**d**) 58TiCuO and (**e**) 75TiCuO, grown with a cationic ration of Cu/Ti = 1.

The AFM images of the TiO₂ _3 WOF (without microflowers) and Ti-Cu-O films are presented in Figure 2. The RMS roughness values thus obtained were 10 ± 1 nm for TiO₂ _3 WOF, 20 ± 3 nm for 16TiCuO, 29 ± 4 nm for 58TiCuO and 33 ± 5 nm for 75TiCuO films.

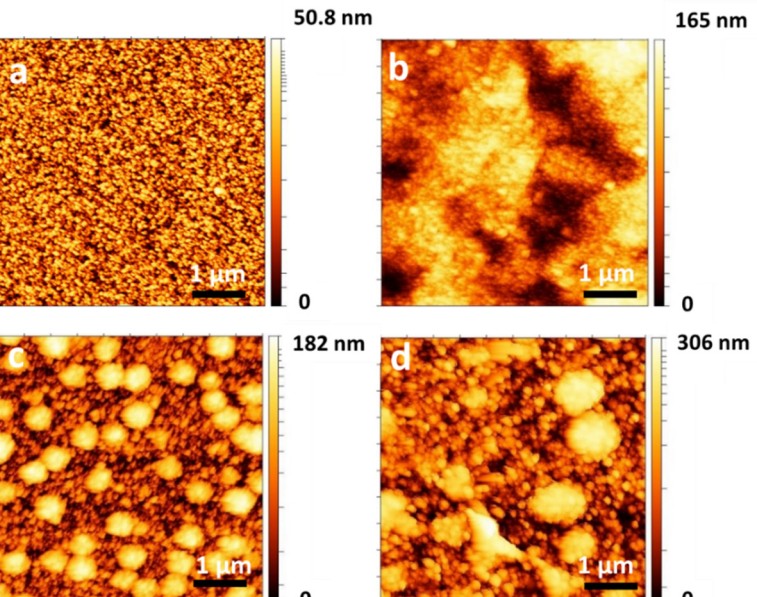

**Figure 2.** AFM images (5 μm × 5 μm) of (**a**) TiO₂_WOF, (**b**) 16TiCuO, (**c**) 58TiCuO and (**d**) 75TiCuO films.

## 3.2. Structure Analysis

The Ti-Cu-O films were analyzed by XRD and Raman spectroscopy to identify the phases present in the layer. Horzum et al. investigated the effect on the properties of TiO₂ films of high concentrations (up to 43 at.%) of Cu. They showed that Cu is incorporated homogeneously into the TiO₂ matrix

until 10 at.%. On the other hand, 20 at.% and 43 at.% Cu-incorporated films presented CuO in their composition [31]. Celik et al. [32] identified the presence of anatase $TiO_2$, CuO and $Cu_4Ti$ phases in Cu-doped films containing 0.007 and 0.18 Cu/Ti molar ratios. The presence of additional $Cu_3TiO_4$ and $Ti_3O_5$ phases was reported for films containing a 0.73 Cu/Ti molar ratio. Saha et al. revealed the presence of anatase $TiO_2$ and $Cu_3TiO_4$ phases in a Cu-Ti composite oxide catalyst presenting 0.5, 1 and 2 Cu/Ti molar ratios [33].

The XRD patterns of the $TiO_2$ film and the Ti-Cu-O films with various Cu contents were acquired in a θ–2θ Bragg–Brentano configuration. The XRD results of the films deposited on the Corning glass substrates are presented in Figure 3. All samples exhibit diffraction peaks of the $TiO_2$ anatase phase as evaluated with JCPDS Card No.21-1272 [34]. The films were polycrystalline, and only diffraction peaks related to the anatase phase were detected in the cases of pure $TiO_2$ and low Cu concentrations (16 at.%). Thin films containing higher Cu percentages (58 and 76 at.%) presented a diffraction peak attributed to $Cu_2O$, which was attested with a peak pointed at 36.43° assigned to the $Cu_2O$ (111) reflection, in agreement with JCPDS card No. 05-0667. To probe the presence of Cu-rich phases, such as $Cu_3TiO_4$, in these two samples, GIXRD was performed to increase the sensitivity of the secondary phase detection. The diffraction patterns thus obtained are shown in Figure 4.

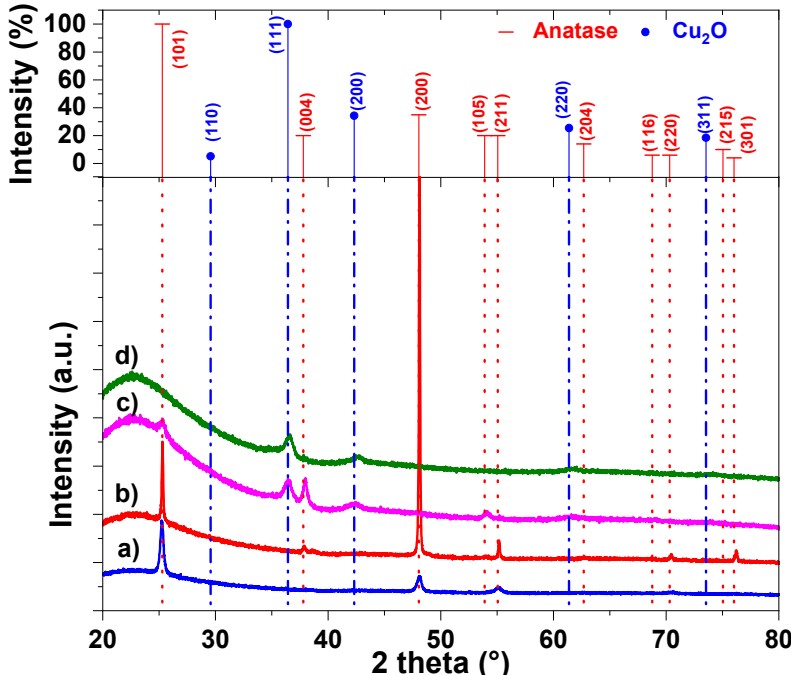

**Figure 3.** XRD diagrams in a Bragg–Brentano configuration of $TiO_2$ and Ti-Cu-O films deposited on Corning glass substrate at 550 °C: (**a**) $TiO_2$, (**b**) 16TiCuO, (**c**) 58TiCuO and (**d**) 76TiCuO. In the upper part of the graph, the theoretical diffraction peaks for $TiO_2$ anatase and $Cu_2O$ correspond to JCPDS Card No.21-1272 and JCPDS card No. 05-0667, respectively.

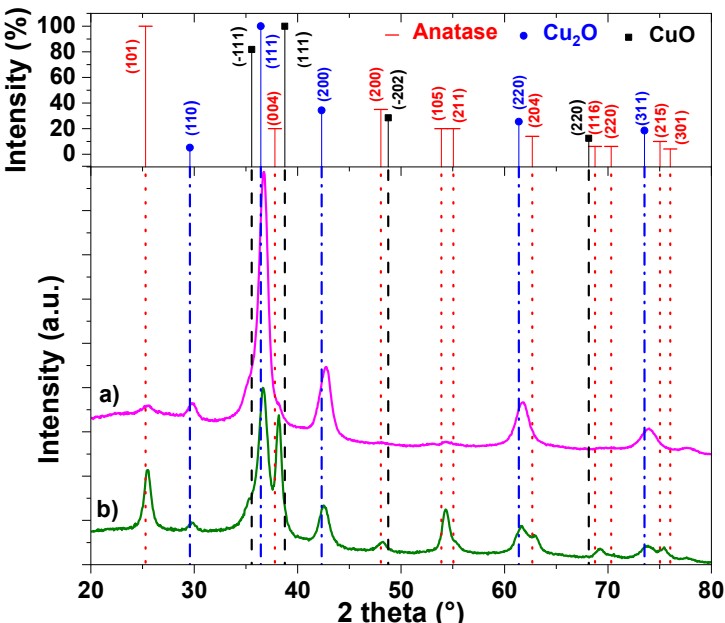

**Figure 4.** XRD diagrams in GI-XRD configuration of TiO$_2$ and Ti-Cu-O films deposited on Corning glass substrate at 550 °C: (**a**) 58TiCuO and (**b**) 76TiCuO. In the upper part of the graph, the theoretical diffraction peaks for TiO$_2$ anatase, Cu$_2$O and CuO correspond to JCPDS Card No.21-1272, JCPDS card No. 05-0667 and JCPDS card No.07-1375, respectively.

The presence of TiO$_2$ anatase and Cu$_2$O phases was corroborated, and also a CuO phase was detected. Indeed, two weak diffraction peaks identified at 2θ = 35.55° and 38.75° in the GIXRD diagrams were assigned to the (−1 1 1) and (1 1 1) CuO reflection lines, respectively, as confirmed by the JCPDS card No.07-1375. No other secondary phase was found in the XRD diagrams, whatever the film composition.

Raman spectroscopy was used to probe the sample in a manner complementary to XRD analysis, working with a more local analysis. Raman spectra of the as-deposited Ti-Cu-O thin films on Corning glass are presented in Figure 5. The following Raman peaks observed for all samples are characteristic of the TiO$_2$ anatase phase: E$_g$ modes at 143, 197 and 639 cm$^{-1}$, B$_{1g}$ mode at 400 cm$^{-1}$, and overlapped A$_{1g}$ and B$_{1g}$ modes at 519 cm$^{-1}$ [35]. It is noticeable that the anatase phase is retained in the Cu-Ti-O films despite the Cu's incorporation into the films, but the anatase peaks are less intense and broadened for films containing more than 50% of Cu, which indicates an increase in structural disorder [36]. Besides, the formation of the Cu$_2$O phase in the two films with Cu content greater than 50 at.% is confirmed in Figure 4 by the presence of several Cu$_2$O Raman modes observed at ~88 cm$^{-1}$, ~106 cm$^{-1}$, ~148 cm$^{-1}$, ~215 cm$^{-1}$ and ~625 cm$^{-1}$ [37,38]. The special feature of the Cu$_2$O Raman spectrum is that only one mode (F$_{2g}$ at 496–515 cm$^{-1}$) is Raman active among the 15 optical modes expected for the cuprite crystal structure [39]. This mode is difficult to observe due to its overlapping with the anatase A$_{1g}$ and B$_{1g}$ modes at 519 cm$^{-1}$. All other modes visible in the Cu$_2$O Raman spectrum are either overtones, combination modes or IR active modes that are allowed to be Raman active through defects and thus relaxations of the selection rules. Finally, the CuO phase is barely detected for Ti-Cu-O films with a Cu content more than 50 at.%, with three weak Raman modes at 298 cm$^{-1}$ (A$_g$), 346 cm$^{-1}$ (B$_g$) and 632 cm$^{-1}$ (B$_g$), characteristic of CuO [40,41].

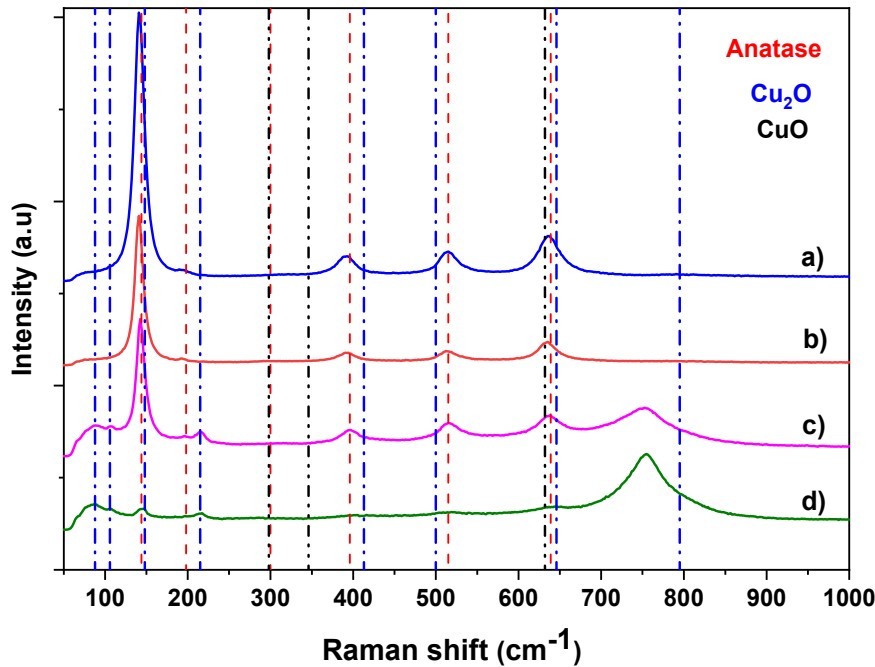

**Figure 5.** Raman spectra of Ti-Cu-O films deposited by aerosol CVD at 550 °C on Corning glass substrates (**a**) $TiO_2$, (**b**) 16TiCuO, (**c**) 58TiCuO and (**d**) 76TiCuO. The dashed lines correspond to the theoretical position of $TiO_2$ anatase, $Cu_2O$ and CuO.

We can conclude that films deposited with a mixed solution of Ti and Cu are mainly composed of $TiO_2$ and $Cu_2O$, but the anatase crystal structure seems to be distorted by the presence of Cu, probably due to its partial incorporation in the crystal network. This distortion can be explained by the fact that the ionic radius of $Cu^+$ (0.77 Å) is slightly larger than that of $Ti^{+4}$ (0.68 Å). Furthermore, because of the difference in charges between $Cu^+$ and $Ti^{+4}$, the addition of $Cu^+$ into the $TiO_2$ matrix probably creates oxygen vacancies inside the lattice in order to maintain the charge neutrality [42], and the higher concentration of $Cu^+$ favors a greater amount of oxygen vacancies in the $TiO_2$ lattice. The increasing concentration of oxygen vacancies with Cu content will undoubtedly contribute to the formation of more and more lattice distortions, which probably explains why all the Raman peaks shift and broaden. Another consequence of copper incorporation could be the occurrence of an additional Raman peak at 755 $cm^{-1}$ in the spectra of Ti-Cu-O films with Cu content greater than 50 at.%, the intensity of which increases with the increase in Cu. The origin of this has remained unexplained until now. However, it is quite surprising that its position exactly corresponds to that of an IR active mode of $TiO_2$ anatase ($A_{2u}$ LO mode) [43,44]. In the paper of Grujić-Brojčin et al. [44], two additional modes were observed in the IR spectrum of $TiO_2$ anatase nanopowders. These have been understood as representing the two strongest $E_g$ Raman modes of anatase being IR-forbidden, and correlated with the presence of oxygen vacancies at the surface of crystalline $TiO_2$ nanoparticles. In our case, the reverse situation could be possible, but this is quite questionable due to the high intensity of the additional Raman line.

The total transmittance of pure $TiO_2$ and Ti-Cu-O films deposited on Corning glass was recorded using an integrating sphere to take into accont the direct and diffusse transmittance. The spectra obtained for the $TiO_2$, 16TiCuO and 75TiCuO films are presented in Figure 6a. The pure $TiO_2$ films are colorless and transparent to the naked eye. Its transmittance exhibits a strong decrease with wavelengths shorter than 390 nm, namely in the UV region. As Cu is incorporated, the films become darker and yellow, and their transmittance decreases and the absorption edge shifts towards longer wavelengths in the visible region. Pure $TiO_2$ thin films are transparent to visible light, with an expected indirect band gap energy of 3.2 eV for the anatase phase [45], thus enabling the absorption of only UV

light with wavelengths shorter than 390 nm; Cu₂O thin films presents an indirect band gap energy of 2.17 eV [46].

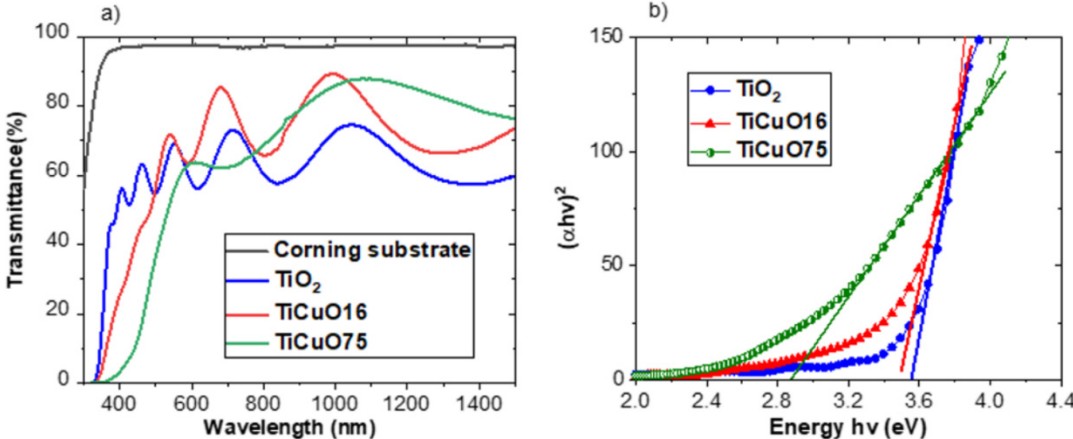

**Figure 6.** (**a**) Total transmittance of Ti-Cu-O films deposited at 550 °C on Corning glass substrates for pure TiO₂, 16TiCuO and 75TiCuO films. (**b**) Tauc plot of the absorbance for the indirect gap as calculated from the transmittance curves of the same Ti-Cu-O films.

The band gap energy of the three films was calculated from their transmittance spectra using the Tauc plot, through the following equation [47]:

$$(\alpha h v)^n = A(h v - E_g)$$

where $\alpha$ is the absorption coefficient, hv is the photon energy (in eV), A is a proportionality constant and $E_g$ is the band gap energy, while n stands for the type of transition, which may be direct or indirect. In our case, the transition was expected to be indirect and, as such, n = 2. The band gap of the Ti-Cu-O films deposited on Corning glass was determined by calculating the value of the intercept of the straight line at $\alpha$ = 0, as deduced from the Tauc plot, i.e., $(\alpha h v)^2$ versus the photon energy. The Tauc representation and curve fittings are shown in Figure 6b. It can be seen that the steep absorption edges shift towards the visible region as the Cu content increases, which indicates a narrowing in the optical band gap.

A gap energy of 3.55 eV is obtained for pure TiO₂. The optical band gap energy shifts towards a visible regime with increasing Cu content, these band gap energies being 3.4 eV for 16TiCuO and 2.88 eV for 75TiCuO. This last value is still higher than the theoretical calculation value of 2.5 eV reported for Cu₂O films [48], probably due to the effect of the mixed TiO₂-Cu₂O structure. Navas et al. [49] reported a large reduction in band gap with increasing copper concentrations, due to the covalent character of the Cu-O interaction leading to new states at the maximum valence band.

*3.3. Photocatalytic Properties*

The measurement and quantification of the absorption band characteristic of Orange G dye at 480 nm allowed us to follow its evolution with altered UV exposure time. This is also a standard way to evaluate the degradation provoked by photoactive materials [50]. Pure TiO₂ films (TiO₂_3 WF) were previously investigated in our last study [26], and used as internal references for this work. Let us remind the reader that the high photocatalytic activity of these films was explained by the formation of microflowers with nanometric petals, which increase the efficient surface and decrease the crystal size. The photocatalytic activity of the Ti-Cu-O films deposited on silicon substrates was evaluated in the same conditions. The degradation kinetics associated with UV irradiation (371 nm) are presented in Figure 7 through the logarithmic variation in the concentration of Orange G normalized to its initial concentration. The slopes of the degradation curves are correlated with the kinetics constants of the

photoactivity. Table 1 presents the evolution of the k constant for Orange G degradation obtained from these results.

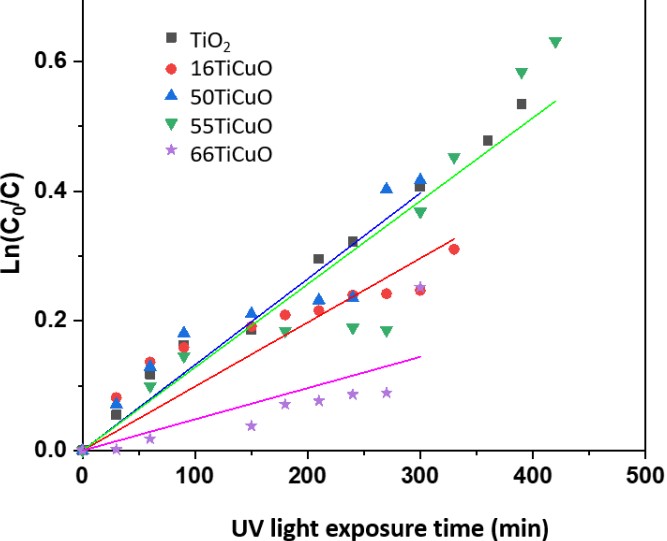

**Figure 7.** Kinetics of Orange G degradation by the $TiO_2$ and Ti-Cu-O films with different Cu content under UV light (371 nm) irradiation. The degradation is calculated from the Orange G concentration measured by the absorption band at 480 nm. The lines correspond to the linear fitting of the curve related to the kinetics constant of the photoactivity.

**Table 1.** Evolution of the k constant for Orange G degradation calculated from the slopes in Figure 7 as a function of copper content.

| Cu (at.%) | k (min$^{-1}$) |
|---|---|
| 0 | 0.00136 |
| 16 | 0.00110 |
| 50 | 0.00132 |
| 55 | 0.00128 |
| 66 | 0.00042 |

The values of the kinetics constants of Ti-Cu-O films, related to the photocatalytic decomposition of Orange G dye, were quite similar to that of pure $TiO_2$, except for the film with the highest Cu concentration (66 at.%). The lowest Cu concentration of 16 at.% slightly decreased the photoactivity, but high values of 0.00132 and 0.00128 min$^{-1}$ were obtained for 50TiCuO and 55TiCuO films. These results indicate that there is an optimal value for the copper content, and that a greater content of Cu is detrimental to the photocatalytic activity in these experimental conditions. Taking into account the possible experimental errors, we consider that Ti-Cu-O films show a photocatalytic response comparable to our best $TiO_2$ film for an UV excitation of 371 nm. Even considering that the kinetics constant values are comparable, we have to take into account the microflower structure of pure $TiO_2$ films resulting in a higher efficient surface for interaction with Orange G dye. Indeed, this microflower structure leads to a higher decomposition rate of Orange G under UV light irradiation than $TiO_2$ without microflowers [26].

A large number of studies have reported that the addition of dissolved copper ions to a $TiO_2$ reaction system improves considerably the rate of photocatalytic oxidation [51–53]. A similar result was found in Cu-doped $TiO_2$ deposited by sputtering [15]. Morikawa et al. reported [54] that loading with copper on $TiO_2$ particles can lead to a photocatalytic activity twice as high as that of pure $TiO_2$. Bard et al. [55] reported that $Cu^{2+}$ ions retarded the electron–hole recombination by the trapping of photogenerated electrons.

The present work shows that even if Ti-Cu-O films have a smaller surface area, they can reach a decomposition rate of Orange G equal to or greater than that of $TiO_2$ films with microflowers under UV. It is important to highlight that since Ti-Cu-O films present a lower band gap than $TiO_2$ films, they are expected to work even better when exposed to visible light (390 nm–750 nm). Further studies evaluating the photocatalytical activity of Ti-Cu-O films working with different wavelengths are currently under progress.

Nevertheless, Okamoto et al. [56] found an inhibition of phenol photo-oxidation with a high concentration of $Cu^{2+}$ (50 mM). This can be explained by the fact that the charge-recombination rate increases with the increase in the amount of Cu. Furthermore, these phenomena may be related to the formation of a recombination center of CuO, which accelerates the recombination of electron–hole pairs because the conduction band of ($CuO/Cu_2O$) is lower than that of $TiO_2$. As such, the excited electrons in $TiO_2$ will be inclined to transfer to CuO, instead of staying in $TiO_2$, to recombine with holes [57]. This last explanation is maybe the most relevant to our case.

### 3.4. Antifouling Activity

The prevention of biofouling is possible through different strategies, including biological, chemical and physical ones [58]. In this study, tests were focused on the potential biocidal activity of copper-containing surfaces (16TiCuO and 58TiCuO) and the anti-adhesion activity of microflowers (three different densities were tested: $TiO_2$_3 WOF, $TiO_2$_3 WF and $TiO_2$_6 WF). After the ban of organotin compounds as biocides in antifouling paints, copper has been the most important alternative for decades [59–61]. Creating various topographies is also a strategy for stopping the colonization of microorganisms [62].

Field tests and in vitro tests were performed to investigate the potency of microflower features and copper incorporation. Diatoms were the main phylum observed on all kinds of substrates in natural seawater [62]. Hence, two model diatoms are studied, *Phaeodactylum tricornutum* and *Navicula perminuta*, to understand how topography and surface chemistry influence microalgal colonization.

### 3.4.1. Marine Biofouling Field Tests

Field tests led to the obtaining of representative and preliminary results on antifouling activities. They allow the selection of effective antifouling surfaces for further in vitro tests whose conditions can be controlled. The antifouling activities of $TiO_2$_3 WF, 16TiCuO and 58TiCuOfilms deposited on Corning glass substrates were investigated by immersion in natural seawater, in Kernevel Harbour (Brittany, France, Atlantic Coast). Faÿ et al. [28] reported that Kernevel Harbour could be a representative area that favors biofouling phenomena. Several species, from microfoulers (bacteria, microalgae including mainly diatoms) to macrofoulers (macroalgae, tubeworms, etc.), can be observed on submerged surfaces. In this study, we focused on microfouling with the observation of diatoms.

Figure 8 represents the evolution of the degree of fouling on the three surfaces resulting from 10 to 38 days of field immersion, and the surface coverage (SC) is indicated for each picture. Variations in SC indicate differences in the coatings' performances. After 10 days, a few diatoms were observed on all the coatings. No significant difference in presence diatoms was observed between days 17 and 25 for all the films. However, the 58TiCuO film seemed more covered by diatoms (82% at day 25, compared to 32% and 43% for $TiO_2$_3 WF and 16TiCuO, respectively). After 38 days of immersion, the microflowers of $TiO_2$_3 WF films were still visible. Microfoulers did not settle on them. Furthermore, the persistence of these structures also proves that they are resistant to a seawater environment after 38 days, and are also mechanically stable, which is quite important since microflowers ensure a better photocatalytic activity. Fewer diatoms colonized $TiO_2$_3 WF films in contrast to 58TiCuO films (40% and 95%, respectively). It should be noted that the diatom communities on the three kinds of films seem to be composed of mainly *Amphora*, and a few *Navicula* and *Nitzschia* (based on morphologies). These three genera were previously reported on antifouling coatings [63–66].

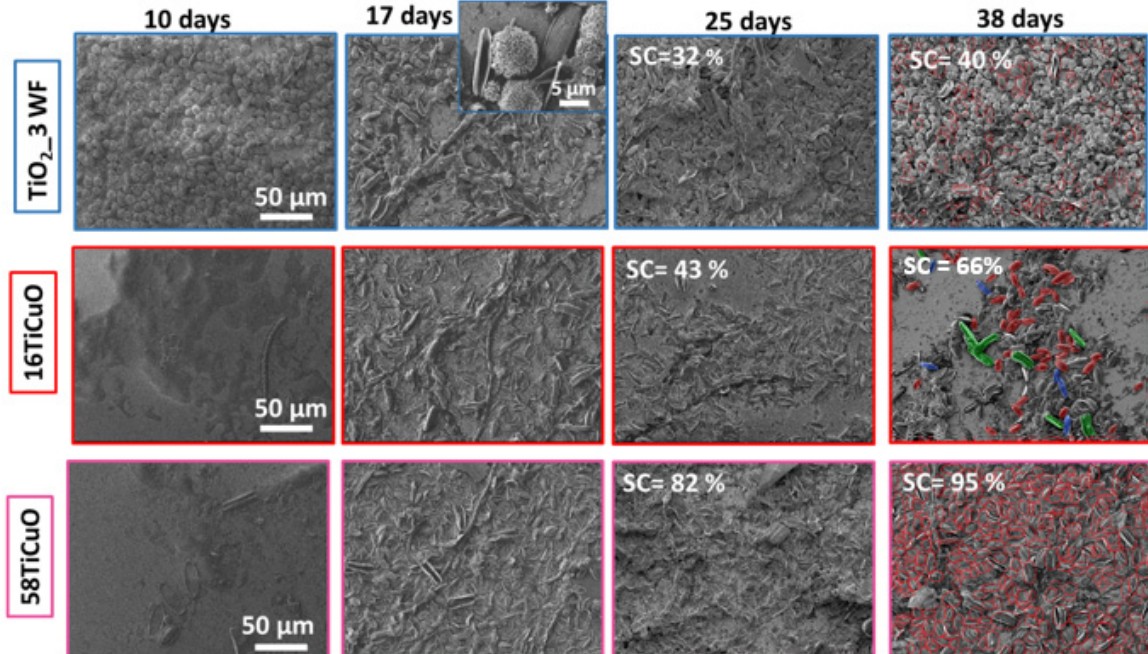

**Figure 8.** Observation by SEM of coatings' colonizations over time until 38 days of immersion in the Atlantic Ocean (Kernevel Harbor, Lorient, France) for TiO$_2$_3 WF, 16TiCuO and 58TiCuO films. The surface coverage (SC) calculated from the image is included for 25 and 38 days. The three main diatom communities are highlighted on the picture corresponding to 16TiCuO after 38 days of immersion; they are identified by their morphology: Amphora (red), Navicula (blue) and Nitzschia (green). For the pictures of TiO$_2$_3 WF and 58TiCuO films after 38 days of immersion, the features highlighted in red correspond to those included in the SC independently of diatom community identification.

The weak adhesion of diatoms on the TiO$_2$_3 WF film could be explained by its morphological topography. Diatoms are known to adhere on rough surfaces whose texture scale is close to a diatom cell's size [23,67,68]. TiO$_2$ coatings present a higher roughness due to their microflower size (Ra = 524 nm) [26], compared to 24–30 nm for Ti-Cu-O films. This morphology could be a limitation to all diatoms settlement. However, the spaces between microflowers and their nanopetals seem to be major parameters. The spaces between microflowers were filled by diatoms whose size was of the same scale as the spaces. The diatoms had probably more difficulty in adhering on the top of the microflowers because of the nanoroughness. This could create obstacles that diatoms could not circumvent to attach strongly onto the micropatterns.

Even if copper has been widely incorporated into marine antifouling paints because of its biocidal activities [60], some microfoulers are copper resistant, such as the diatoms belonging to the *Amphora* genus [65,69,70]. Diatoms are known to cope with toxic metal concentrations. Indeed, they produce extracellular polysaccharides (EPS) to immobilize metals outside the cells, such as copper. As a result, the excretion of these protective products reduces sensitivity to copper exposure [71–73]. So the more copper there is (up to a certain point whereat the concentration of metal is too high to be countered), the more EPS there are, and the more diatoms adhere strongly to the substratum. This explains why lots of *Amphora* are encountered on Ti-Cu-O films. The highest fouling rate was observed on the 58TiCuO film.

### 3.4.2. Marine Biofouling In Vitro Tests

In vitro tests were performed to understand how surfaces influence the adhesion of diatoms under controlled conditions. The two microalgae (*P. tricornutum* and *N. perminuta*) were chosen because they are model organisms in adhesion studies, they are morphologically different [74,75], and *N. perminuta* is from one of the main diatom communities observed in our in situ test. Indeed, *P. tricornutum*

(6.3 ± 0.5 µm long and 2.3 ± 0.2 µm wide, mean ± SD for n = 10) is smaller than *N. perminuta* (8.5 ± 1.7 µm long and 3.0 ± 0.3 µm wide, mean ± SD for n = 10). Moreover, the cell plasticity of the former, conferred by its low silica content frustule when adhered, could allow it to fit more easily to the microtopography [76].

To study the impact of microtopography, surfaces with the same chemical compositions but different microflowers densities were tested in situ with regard to the two diatoms: two surfaces with different microflowers densities, $TiO_2\_3$ WF and $TiO_2\_6$ WF, and the same surface without microflowers, $TiO_2\_3$ WOF [24]. Table 2 shows that the average microflower density of $TiO_2\_3$ WF was 14%, while for the $TiO_2\_6$ WF film it was 22%. The spaces between microflowers for $TiO_2\_3$ WF and $TiO_2\_6$ WF were 15 ± 7 and 8 ± 5 µm long, respectively.

**Table 2.** Average microflower density and space between microflowers for $TiO_2\_3$ WF and $TiO_2\_6$ WF, determined by ImageJ from the SEM micrographies.

| Samples | Average Microflowers Density (%) | Space between Microflowers (µm) |
|---|---|---|
| $TiO_2\_3$ WOF | - | - |
| $TiO_2\_3$ WF | 14 | 15 ± 7 |
| $TiO_2\_6$ WF | 22 | 8 ± 5 |

*P. tricornutum* seemed to adhere more on coatings with higher microflower density ($TiO_2\_6$ WF) than on $TiO_2$ without microflowers ($TiO_2\_3$ WOF) (Figure 9a), with 10.9 ± 1.9% and 9.2 ± 1.8% covering rates, respectively ($p = 0.0275$). However, statistics tests showed no significant difference between the surface without microflowers ($TiO_2\_3$ WOF, 9.2 ± 1.8% covering rates) and the surface with small microflower density ($TiO_2\_3$ WF, 9.0 ± 3.1% covering rates), and a slight difference between both surfaces with microflowers ($TiO_2\_3$ WF and $TiO_2\_6$ WF). As regards the results for $TiO_2\_3$ WOF and $TiO_2\_3$ WF ($p = 0.8884$), and for $TiO_2\_3$ WF and $TiO_2\_6$ WF ($p = 0.1347$), they are statistically the same. We can assume by syllogism that the difference between $TiO_2\_3$ WOF and $TiO_2\_6$ WF is not noteworthy ($p = 0.0275$). This is probably due to the inhomogeneous spaces between microflowers, indicated by the high standard deviations (7 and 5 µm for $TiO_2\_3$ WF and $TiO_2\_6$ WF, respectively). In these conditions, we can conclude that the presence of microflowers at these densities does not significantly affect the adhesion of *P. tricornutum* in spaces between features.

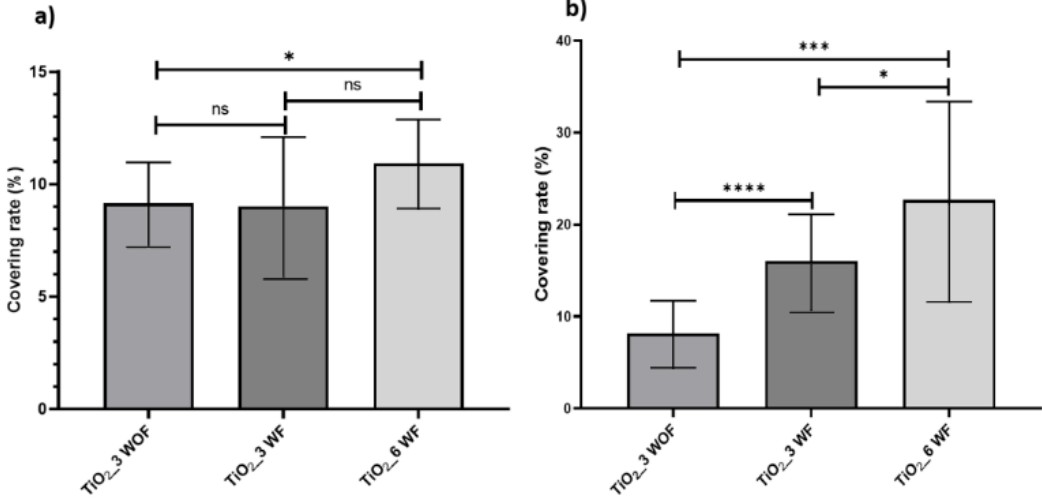

**Figure 9.** Covering rate of (**a**) P. *tricornutum* and (**b**) N. *perminuta* on $TiO_2$ surfaces with different morphologies after 72-h adhesion at 20 °C. The results obtained from the *t*-test are included using the following symbols. ns (not significant): $p > 0.05$; *: $p \leq 0.05$; ***: $p \leq 0.001$; ****: $p \leq 0.0001$ (number of replicates = 18). The standard deviation is represented as bar error.

The density of microflowers impacted *N. perminuta* more than *P. tricornutum* (Figure 9b). Indeed, the higher the density was, the more adhesion occurred: 8.1% ± 3.6%, 16.0% ± 5.1% and 21.0% ± 3.5% for TiO$_2$_3 WOF, TiO$_2$_3 WF and TiO$_2$_6 WF samples, respectively. This is probably due to the roughness and possible attachment points [22,68]. The higher the microflowers density is, the smaller are the spaces between microflowers, and the higher is the number of attachment points (compared to smooth surfaces). It has been proven that the implications of the microtopographies were visible when the features sizes were close to the width of the diatoms' cells [23,67].

In this study, it was found that the size of the spaces between microflowers influences the adhesion of the diatoms, schematically represented in Figure 10. On TiO$_2$_3 WF films, the spaces were about 15 ± 7 µm, while on TiO$_2$_6 WF films, they were about 8 ± 5 µm. *N. perminuta* cells were 8.5 ± 1.7 µm long, and the *P. tricornutum* cells were 6.3 ± 0.5 µm long. TiO$_2$_3 WOF was smooth (R$_{RMS}$ = 10 nm) (Figure 10a,d), and the spaces in TiO$_2$_3 WF films (Figure 10b,e) were almost twice as big as the *P. tricornutum* and *N. perminuta* average cell size. As a result, the spaces on this latest surface were too big to allow both diatoms' cells to attach to two adjacent microflowers, a configuration offering more attachment points and so a better adhesion. The same situation occurred for *P. tricornutum* on TiO$_2$_6 WF: the diatom could not attach to the two neighboring microflowers at the same time, as its cells are too small to fill the spaces completely (Figure 10c). It would be interesting to study the adhesion of these species on TiO$_2$ films with higher microflower density, and with spaces of about 6 µm and less (this is the *P. tricornutum* cells' average size, and so higher covering rates would be expected). On the contrary, the spaces in TiO$_2$_6 WF films were in the same size range as the size of *N. perminuta*. Therefore, it can fit within the spaces, and adhere on at least two adjacent microflowers (Figure 10f).

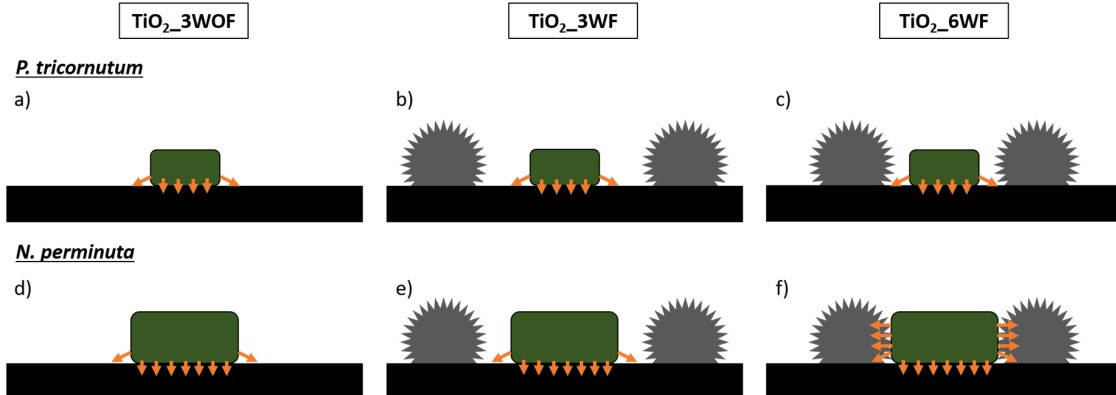

**Figure 10.** Schematic representation of the attachment points (orange arrow) of diatoms (green parallelogram) on surfaces (black rectangle) with microflowers (grey concave polygon). (**a**) P. *tricornutum* on smooth surface (TiO$_2$_3 WOF), (**b**) and (**c**) P. *tricornutum* on surfaces with spaces between features bigger than cells' sizes (TiO$_2$_3 WF and TiO$_2$_6 WF, respectively), (**d**) N. *perminuta* on smooth surface (TiO$_2$_3 WOF), (**e**) N. *perminuta* on a surface with spaces between features bigger than cells' sizes (TiO$_2$_3 WF), (**f**) N. *perminuta* on a surface with spaces between features of the same size as cells (TiO$_2$_6 WF).

The antifouling activity of 16TiCuO films was also investigated on both diatoms because these films exhibited good results in field tests. Covering rates of 9.2 ± 1.8% and 4.9 ± 1.3% were observed for *P. tricornutum* on TiO$_2$_3 WOF and 16TiCuO films, respectively (Figure 11a). The adhesion of this diatom was inhibited by 47% on 16TiCuO films. The composition of these smooth surfaces has an impact on the adhesion of this microorganism. Conversely, no effect of 16TiCuO films on *N. perminuta* adhesion was observed (Figure 11b).

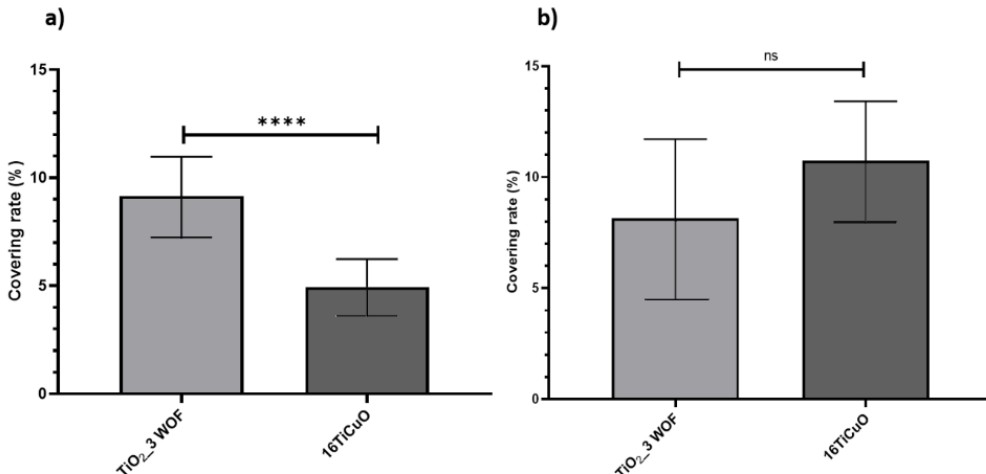

**Figure 11.** Covering rate of (**a**) P. *tricornutum* and (**b**) N. *perminuta* on $TiO_2\_3$ WFO and 16TiCuO surfaces after 72-h adhesion at 20 °C. The results obtained from the *t*-test are included using the following symbols. ns (not significant): $p > 0.05$; ****: $p \leq 0.0001$ (number of replicates = 18). The standard deviation is represented as bar error.

As copper is known as a biocidal metal used in antifouling paints, its toxicity towards both diatoms was investigated with the Sytox Green staining method. No toxicity from the 16TiCuO films was observed towards the planktonic cells of *P. tricornutum* after 72 h of incubation, compared to glass which was used as a non-toxic standard. However, a growth inhibition of 39% was observed for planktonic cells compared to glass ($1.81 \times 10^6$ cells/mL and $2.98 \times 10^6$ cells/mL, respectively). This effect on growth can explain the smaller adhesion rate on 16TiCuO, because fewer cells were present to adhere. Several studies have demonstrated that copper inhibits growth for some diatoms species [77–79]. Copper even blocks the progress between phases G2 and M in the cell cycle for *P. tricornutum*, resulting in non-dividing cells [80,81]. Under our conditions of study, copper was probably not sufficiently released in the water to totally inhibit the division of all the cells, but was enough to decrease the growth rate of the culture without toxicity. A dosage of released $Cu^{2+}$ would be an interesting angle from which to evaluate how 16TiCuO films affect *P. tricornutum*.

Concerning *N. perminuta*, no effect on planktonic cells has been quantified. This diatom sediments more than *P. tricornutum*, and all inoculated cells settled. As a result, the toxicity effect on adhered cells was studied. This diatom was not affected by surface composition ($TiO_2\_3$ WOF nor 16TiCuO) as no significant difference was observed. This is probably due to its detoxification mechanism. This microalgae can intracellularly accumulate copper in some kind of granules, which maintain a low cytoplasmic concentration of copper and so limit toxicity [82]. In these conditions, *N. perminuta* adhered whatever the surface Cu-content. It also has to be noted that both surfaces ($TiO_2\_3$ WOF and 16TiCuO) were smooth ($R_{RMS}$ of 10 and 23.77 nm, respectively), and smooth surfaces do not allow great adhesion of the diatom, as demonstrated previously.

The in vitro tests showed that surface topographies and chemical compositions can select diatom species depending on their morphology and chemical defenses against toxic metal such as copper. *P. tricornutum* was not impacted by the topography in this study as the spaces between its features were bigger than the cells' sizes in all cases. The study of the adhesion of *N. perminuta* showed that spaces in the same size range as cells allowed better attachment rates, while smooth surfaces inhibit adhesion. The results are consistent with other studies, which illustrates well the attachment point theory whereby topography scale, referring to spaces between features, influences the adhesion of microorganisms [23]. In our study, the larger the spaces between microflowers were, the more diatoms could move into them, and the more difficult it was for them to choose where to adhere. On the contrary, as observed for *N. perminuta*, the smaller the spaces, the better it adhered.

Concerning the chemical compositions of the surfaces, the two diatoms had different behaviors: *P. tricornutum* was sensitive to copper, and its growth rate was impacted by this metal in a manner contrary to *N. perminuta*. This diatom developed a strategy to counter copper's toxic effects. This explains why *Navicula* cells, among the numerous *Amphora* cells, can be observed in field tests the same as in in vitro tests.

## 4. Conclusions

In order to enhance the properties of the $TiO_2$ wide band gap semiconductor, new Ti-Cu-O films with high contents of copper (16 to 75 at.%) were deposited by AAMOCVD at 550 °C. These thin films were proposed as an efficient visible light-active photocatalytic material, presenting a high surface area for the adsorption of targeted species and a good chemical stability in water.

Ti-Cu-O films were composed of $TiO_2$ anatase, $Cu_2O$, and CuO with Cu cationic content higher than 50 at.%. Besides this, a band gap decrease was observed when increasing the Cu content in Ti-Cu-O films, from 3.55 eV with pure $TiO_2$ to the lowest value close to 2.88 eV with the highest Cu content (75 at.% Cu). These films are very promising materials for the photocatalytic degradation of organic compounds under visible light. They presented a maximum degradation kinetics constant value of 0.00132 $min^{-1}$ when decomposing Orange G dye, equivalent to that of pure $TiO_2$ with microflowers.

The antifouling tests of $TiO_2$ and Ti-Cu-O films are also promising. Marine biofouling field tests in Lorient's Harbor in France showed a reduction in colonization for $TiO_2$ and 16TiCuO films after 38 days of immersion. The size of the microorganisms should be taken into consideration. In vitro tests using two diatoms (*P. tricornutum* and *N. perminuta*) showed that spaces between microflowers play a significant role in the adhesion of diatoms: microalgae adhere less when spaces are bigger than their cells, compared to when spaces are of the same size as their cells. Indeed, this study confirms the attachment point theory: diatoms of sizes similar to these spaces could adhere within the spaces. The topography and chemical composition can act differently depending on the organism's morphology and the presence of mechanisms of defense.

Films containing Cu did not alter *N. perminuta* growth or adhesion, while they affected *P. tricornutum* by lowering its growth rate and adhesion without noticeable toxicity. Indeed, Cu-Ti-O is a very promising non-toxic fouling release film for marine and industrial applications.

Developing non-toxic foul-release films for marine and industrial applications is the particular appeal of this work. Future research could focus on gathering the best properties from each film (chemical composition, topography) and combining them to produce an effective antifouling surface resistant to a variety of foulers of various trophic scales.

**Author Contributions:** Authors in order of appearance in the manuscript. C.V.d.O.: Films deposition and physico-chemical characterization, writing (Original Draft Preparation). J.P.: Performed and analyzed the biofouling experiments, interpreted data, wrote the final version of the manuscript. F.F.: designed the biofouling experiments and interpreted data, wrote the final version of the manuscript. F.S. (Frédéric Sanchette): Supervision, discussion and article correction. F.S. (Frédéric Schuster): Funding acquisition. A.A.: Supervision and article correction. O.C.-P.: performed and analyzed the Raman experiments. J.-L.D.: Help in AACVD and optical characterization. C.J.: conceived and designed the experiments, analyzed data and wrote the final version of the manuscript. All authors have read and agreed to the published version of manuscript.

**Funding:** This work benefced of the financial help of the ANR through the Centre of Excellence of Multifunctional Architectured Materials "CEMAM" no. ANR-10-LABX-44-01 and REACT-PIRE "ANR-15-PIRE-0001" It was also co-funded by the GIP52 (Groupement d'Intérêt Public Haute-Marne) and CEA Saclay.

**Acknowledgments:** We thanks H. Roussel for the XRD experiments and D. Riasseto for the fruitful discussion on photo-catalysis experiments.

**Conflicts of Interest:** The authors declare no conflict of interest.

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
