# Peer review of "Marine Antibiofouling Properties of TiO2 and Ti-Cu-O Films Deposited by Aerosol-Assisted Chemical Vapor Deposition"

_coatings, doi:10.3390/coatings10080779_

Round 1

Reviewer 1 Report

Caroline et al. reported the “Marine antibiofouling properties of TiO2 and Ti-Cu-O films deposited by Aerosol-Assisted Chemical Vapor Deposition”. The authors have characterized the sample extensively to achieve the desired properties. A few more clarifications are, however, needed before the manuscript can be considered suitable for publication.  A list of other comments that need to be addressed follows:

  1. An introduction is poorly written; please include the scope and limitations of the TiO2 microparticles, their advantages, and disadvantages since TiO2 is ubiquitous material, and it is hard to find the novelty in the paper. Please justify the novelty in the introduction.
  2. The first paragraph contains trivial statements. The introduction should be reduced in length and have a focus on current analytical challenges. Essential related works can be cited
  3. Experimental part: design of experiments is missing like why authors choose specific parameters without optimization?
  4. Figure 3: pleas improve the quality of the XRD plot by including the JCPDS data card number.
  5. Include these recent articles and compare the results with them by including a comparison table: Surface & Coatings Technology 382 (2020) 125154, ACS Appl. Mater. Interfaces 2018, 10, 40, 34087–34097, Catal. Sci. Technol., 2019,9, 652-658, Nanomaterials 2020, 10(5), 882
  6. The quality of some figures is inferior and needs to be enhanced.
  7. It is better to check and correct the font size of the x and y-axis of all figures in the manuscript. It should be the same.
  8. Please include error-bar for Figure 11

Author Response

Reviewer #1

We thank to the referee for the critical revision of our manuscript

Comments and Suggestions for Authors

Caroline et al. reported the “Marine antibiofouling properties of TiO2 and Ti-Cu-O films deposited by Aerosol-Assisted Chemical Vapor Deposition”. The authors have characterized the sample extensively to achieve the desired properties. A few more clarifications are, however, needed before the manuscript can be considered suitable for publication.  A list of other comments that need to be addressed follows:

  1. An introduction is poorly written; please include the scope and limitations of the TiO2microparticles, their advantages, and disadvantages since TiO2 is ubiquitous material, and it is hard to find the novelty in the paper. Please justify the novelty in the introduction.
  2. The first paragraph contains trivial statements. The introduction should be reduced in length and have a focus on current analytical challenges. Essential related works can be cited
  3. it is hard to find the novelty in the paper. Please justify the novelty in the introduction.

To answer to these 3 issues, we have modified the introduction to explain better the novelty of the work and to include the suggested reference (lines 40-88)

4. Experimental part: design of experiments is missing like why authors choose specific parameters without optimization?

We have included some sentences to explain the choice of deposition parameters used in this work (line 194-197)

“The optimization of TiO2 films deposition was already presented in [20]; we used the same deposition parameter is this study for TiO2 and Cu-Ti-O films, mainly characterized by a deposition temperature of 550 °C for 40 minutes. A carrier gas flow of 6 L.min-1 and a solution feeding rate of 3 mL.min-1 were used.

5. Figure 3: please improve the quality of the XRD plot by including the JCPDS data card number.

We have included in the figure captions of Figure 3 and 4 the JCPDS data card number as suggested

6. Include these recent articles and compare the results with them by including a comparison table: Surface & Coatings Technology 382 (2020) 125154, ACS Appl. Mater. Interfaces 2018, 10, 40, 34087–34097, Catal. Sci. Technol., 2019,9, 652-658, Nanomaterials 2020, 10(5), 882

The common point of the suggested references are the photocathalytic activity of TiO2-based compound or WO3. These articles present measurements performed by using different methods. We have interpreted the referee’s requirement as a table comparing the photocathalytic activity of these experiments. Nevertheless, this kind of experiment is not normalised and the comparison of activities cannot be done when using different organic compounds or even different experimental setup. We have included some of the reference as suggested, but we have not included the table because we don’t think it will add value to the work described in our manuscript.

7. The quality of some figures is inferior and needs to be enhanced.

We have tried to enhance the quality, if it is not enough, please indicate the figure we need to rework

8. It is better to check and correct the font size of the x and y-axis of all figures in the manuscript. It should be the same.

We have used the same font, but the final size depends on the resizing of figures

9. Please include error-bar for Figure 11

The error bars are included in the graphs presented in Figure 9 and 11.

Reviewer 2 Report

In the present work, the authors synthesized TiO2 and Ti-Cu-O films by aerosol assisted chemical vapor deposition for marine antibiofouling applications. In my opinion, the experimental work is well-designed and the manuscript is well-written. The authors comprehensively characterized their deposited films employing variety of techniques including XRD, SEM, AFM, EDX, Raman and UV-Vis transmittance spectroscopies, studied photocatalytic activity and performed biofouling and toxicity tests. I think, this work can be accepted for publication after a minor revision. Few comments:

-The entire text must be checked for typos and small mistakes. The sentence located at lines 231-233 must be re-written. Lines 264 and 307 must be checked.

-The authors explain a mismatch between nominal and actual chemical composition of the films by the copper diffusion in vapor phase. However, how can be explained an opposite effect for [Cu]/[Ti] =1/3 and [Cu]/[Ti] =1 films? In the first case actual Cu content is lower, but in the second case it is higher.

-In experimental part the authors say that for AFM analysis several areas were scanned, however they provide only average roughness of the films without errors. I would suggest to add errors.

-Figure 6b. Tangent line for TiCuO16 does not look drawn properly.

- in Conclusions the authors must mention an effect of Cu addition on antifouling properties compared to pristine TiO2 films.

Author Response

Referee # 2

We thank the referee for the carefully reading of our manuscript

“Comments and Suggestions for Authors

In the present work, the authors synthesized TiO2 and Ti-Cu-O films by aerosol assisted chemical vapor deposition for marine antibiofouling applications. In my opinion, the experimental work is well-designed and the manuscript is well-written. The authors comprehensively characterized their deposited films employing variety of techniques including XRD, SEM, AFM, EDX, Raman and UV-Vis transmittance spectroscopies, studied photocatalytic activity and performed biofouling and toxicity tests. I think, this work can be accepted for publication after a minor revision. Few comments:

-The entire text must be checked for typos and small mistakes. The sentence located at lines 231-233 must be re-written. Lines 264 and 307 must be checked.”

We have checked the text and corrected the indicated lines

“-The authors explain a mismatch between nominal and actual chemical composition of the films by the copper diffusion in vapor phase. However, how can be explained an opposite effect for [Cu]/[Ti] =1/3 and [Cu]/[Ti] =1 films? In the first case actual Cu content is lower, but in the second case it is higher.”
In fact, Cu richer solutions lead to Cu rich layers. The difference in cationic content of the films is 16% for a nominal value of 25% in the solution. This deviation of composition is quite usual in MOCVD, probably due to a difference in the deposition rate if each element.

Nevertheless, when using the Cu]/[Ti] =1, not only the cationic concentration is different, but the total cationic concentration is higher (0.03M instead 0.02M). In this case, the composition is homogeneous within the deposited films, but the run to run variability is higher. This lack in reproducibility appears only for Cu rich solutions. As pure Cu solution didn’t succeed in growing a continuous film, there should be a cooperative effect of the two cations during deposition, not explained until now. Further studies are currently performed to identify the deposition parameters that control the kinetics.

We have modified the paragraph between lines 326-335

“-In experimental part the authors say that for AFM analysis several areas were scanned, however they provide only average roughness of the films without errors. I would suggest to add errors.”

We have included the error in the rms roughness values.( line 388-389)

“-Figure 6b. Tangent line for TiCuO16 does not look drawn properly.”

We have corrected this mistake, thanks for checking

“- in Conclusions the authors must mention an effect of Cu addition on antifouling properties compared to pristine TiO2 films.”

We have modified the conclusions to highlight the effect of Cu addition on antifouling propertie ‘lines 779-794.

Reviewer 3 Report

The authors have deposited TiO2 and Ti-Cu-O films via aerosol-assisted MOCVD method. These films are introduced as a visible light sensitizer catalyst. It is well described manuscript which is suitable for a publication in Coatings after minor corrections:

  • What is beneficial of use of Cu atoms in TiO2 instead of Mn, Ni and Fe? Is the photocatalyst (band gaps) more ideal when doping with Cu?
  • Typo line 114: aerosol-assisted instead of aerosol assisted.
  • Section 2.5.4 is not clear to me. Can the authors improve this section in case the readers will understand how data were analyzed
  • Line 283-286: The authors have mentioned the deposition was done in previous work, but a briefly explanation why some specific parameters were introduced is welcome. For example: “when working with very high precursor flow rate”, why?
  • Line 286-287: “very high” and “high specific” is not scientific correct, please use value. The readers need to know which values are “very high”
  • Line 290: Why is the thickness homogeneity not optimal?
  • The sample preparation for cross-sectional SEM measurements was not described in experimental section.
  • Line 323: which glass?
  • Figure 11: These graphs show lack of details as the readers will not understand these data analysis.
  • Line 779: What is AAMOCDVD as this abbreviation is not introduced.
  • How sure is the authors that formation of Cu2O and CuO phase in your film occurred? Have the authors also considered the measurements of a valence state of Cu?

Author Response

 We thank the reviewer for the comments. We have included some modifications in the manuscript as requested and we answer in the joined file to some comments.

Round 2

Reviewer 1 Report

Some answers to the comments are not satisfactory, and there is no clear information about why TiO2 since there are plenty of papers. Novelty is missing. Some suggested references are not cited well, and even some references are very old. However, the article requires more refinement before publication.

Author Response

We have checked the manuscript and modified the introduction to add some recent references about works searching to improve the TiO2 photocatalytical activity and its application to self-cleaning surface and antifouling.  Special attention have been paid to the use of Cu in combination with TiO2.   The paragraph explaining the novelty of the work has been also rewritten to highlight the main results of our work.

Round 3

Reviewer 1 Report

All the comments addressed properly, the manuscript is ready for the publication